# Green Space Compactness and Configuration to Reduce Carbon Emissions from Energy Use in Buildings

Ranran Ji [1,2], Kai Wang [3], Mengran Zhou [1], Yun Zhang [4], Yujia Bai [1,5], Xian Wu [1], Han Yan [1,5], Zhuoqun Zhao [1,5] and Hong Ye [1,2,6,*]

1    Key Laboratory of Urban Environment and Health, Institute of Urban Environment, Chinese Academy of Sciences, Xiamen 361021, China
2    CAS Haixi Industrial Technology Innovation Center in Beilun, Ningbo 315830, China
3    China-UK Low Carbon College, Shanghai Jiao Tong University, Shanghai 200240, China
4    School of Computer and Date Science, Xiamen University Malaysia, Sepang 43900, Malaysia
5    University of Chinese Academy of Sciences, Beijing 100049, China
6    Xiamen Key Laboratory of Urban Metabolism, Xiamen 361021, China
*    Correspondence: hye@iue.ac.cn

**Abstract:** Building sector consists of a major part of global energy consumption and carbon emission. Reducing energy consumption in buildings can make a substantial contribution towards the strategic goal of carbon neutrality. Building energy consumption carbon emission (BECCE) is highly correlated with microclimate. Green space has long been recognized as the natural way to improve the microclimate and reduce BECCE. However, the effective distance and optimized configuration of green space for the reduction in BECCE are hardly known. To this purpose, we developed a green space compactness (GSC) index as an indicator of microclimate around the People's Bank, located in 59 cities across China, and used statistical, deep learning, and spatial analysis methods to obtain the most effective distance with respect to the effect of GSC on BECCE. We used hot and cold spot spatial analysis methods to detect the spatial heterogeneity of BECCE and analyzed the corresponding GCS to discover the optimal way for BECCE reduction. The results clearly showed that BECCE was highly correlated with the GSC, and the influence of GSC on BECCE was the highest at the distance of 250 m from the building. The hot and cold spots analysis suggested that BECCE has a significant spatial heterogeneity, which was much higher in the north part of China. Improving the configuration of green space for certain cities could lead to considerable emission reductions. If the BEECE is reduced from 4675 tons to 486 tons, the GSC needs to be increased from 0.39 to 0.56. The study suggests that 250 m is the most effective distance to reduce BECCE, and optimal green space configuration can provide a feasible way to mitigate carbon emissions and valuable information for the development of low-carbon cities.

**Keywords:** urban green space compactness index; microclimate; building energy consumption; peak carbon; carbon neutrality

## 1. Introduction

Global warming is the most important environmental issue and the most complex challenge of the 21st century [1,2]. According to the Sixth Assessment Report of the Intergovernmental Panel on Climate Change (IPCC), global surface temperature has increased by 1.09 °C between 1850–1900 and 2011–2020 [3]. To address the severe consequences of global warming, China has proposed to reach peak $CO_2$ emissions by 2030 and be carbon neutral by 2060 [4,5]. As urbanization continues at an unprecedented rate and scale, carbon emissions continue to increase and exacerbate global warming [6]. Although urban areas cover only approximately 2% of the Earth's surface, they are the source of more than 80% of the world's carbon emissions [7,8]. Buildings occupy a relatively large part of the urban area. The building sector has an important role in combating global $CO_2$ warming [9]. In

recent years, energy use in the building sector in China has continued to grow. In 2019, total carbon emissions in China were approximately 10 billion tons; the building sector emitted 3.98 billion tons of $CO_2$. Without strong and effective emission reduction policies, the contribution of energy use in buildings in China to total global emissions in 2050 is likely to be substantial [10]. Therefore, reductions in energy use in the building sector can contribute to effective mitigation of carbon emissions [11].

The impact of green space on urban building energy consumption is mainly realized by its cooling and shading effects. The cooling effect of green space alleviates the well-known urban heat island (UHI) phenomenon [12–14]. Numerous studies have shown that the configuration of urban green spaces indirectly affects microclimate by influencing urban ecology and reducing the urban heat island effect [15–19]. Previous studies show that building energy consumption carbon emission (BECCE) is closely related to the building itself, the surrounding natural environment, and the socioeconomic context [20]. The natural environment includes regional climate and microclimate. The effect of microclimate on BECCE is approximately 10%. However, microclimate can easily be controlled and modified. Therefore, microclimate improvement is an integral part of reducing BECCE [21,22].

There are few studies on the effective distance of green space for BECCE and optimal configuration of green space, and research on this topic will contribute to the development of low-carbon cities. Some attempts have been made to explore the relationship between carbon emissions and drivers through linear regression, machine learning, and spatial analysis. For example, Shen et al. [23] established a partial least squares (PLS) model to identify the crucial effects and pathways affecting carbon emissions. Some machine learning algorithms such as Random Forest have been successfully applied to estimate population, which is relevant to carbon emissions [24]. Xu et al. [25] used geographical detector model (GDM) to explore the driving forces of carbon dioxide emissions in China's cities. We examined the office buildings of the People's Bank of China in 59 cities across China. Each building was placed in the center of a grid; buffer zones were drawn around the building; 20 buffer zones extended across a total of 1 km along each direction of the x and y axes of the grid. A green space compactness index was developed to characterize the microclimate in the buffer zones. Partial least squares regression, random forest and geographical detection models were constructed using seven factors influencing building emissions to explore the contribution of GSC to BECCE for different buffer distances. We identified the buffer distances that are the most effective for the emission reductions, examined the spatial heterogeneity of BECCE across China, and suggested strategies to optimize green space configuration for emission reduction.

## 2. Data

### 2.1. Remote Sensing Data

The green space data used in this study are based on the world's first 10 m resolution global surface cover product—FROM-GLC10, which was developed by the Gong Peng research group [26].

### 2.2. Socioeconomic Data

In this study, we used the data from 59 office buildings of the People's Bank of China. They were public buildings in 59 cities across the country. These buildings belong to different cities, but they serve the same function. Gross Domestic Product per person (GRP) and education spending (ES) for each building were obtained from 2018 yearly statistical book of the city (http://www.stats.gov.cn/tjsj/ndsj/2018/indexch.htm, accessed on 24 March 2022).

### 2.3. Regional Climate Data

We used 2018 daily average temperature data from the weather station that was closest to each building. The data were provided by the China Meteorological Administration (http://data.cma.cn/ accessed on 4 August 2021). We calculated cooling degree days

(CDD) and heating degree days (HDD) for each station. The CDD is the cumulative number of degrees for which the average daily temperature is above 26 °C, while HDD is the cumulative number of degrees for which the average daily temperature is below 18 °C; they were calculated using the basic formulas expressed in Equations (1) and (2), respectively [27]:

$$CDD26 = \begin{cases} T - 26°C & if \ T > 26°C \\ 0 & if \ T < 26°C \end{cases} \tag{1}$$

$$HDD18 = \begin{cases} 18°C - T & if \ T < 18°C \\ 0 & if \ T > 18°C \end{cases} \tag{2}$$

where $T$ is the average air temperature of a day. Cumulative values for the year were calculated from the sum of daily values.

*2.4. Calculating Carbon Emissions*

The carbon emissions from the building fuel consumption of various office buildings of the People's Bank of China in 2018 were collected, including energy consumption patterns such as electricity, coal, natural gas, and heat. To account for differences in energy type, we converted the consumption of all types of energy into equivalent carbon emissions. This was determined by the amount of fuel consumed and the carbon content of the fuel. The building's carbon emissions were determined by its consumption of different fuels and carbon content. Equation (3) gives the framework for converting energy use into carbon emissions. This calculation to estimate residential emissions is based on the Intergovernmental Panel on Climate Change [28] formula for national greenhouse gas emissions:

$$CE_F = \sum_{i}^{n} (C_{ei} \times G_{ei} \times I_{ei}) \tag{3}$$

where $CE_F$ represents carbon emissions from fuel consumption, $C_{ei}$ is the consumption of fuel $i$, $G_{ei}$ is the energy conversion factor of fuel $i$ (MJ/Tt or MJ/Mm$^3$), which is 111 MJ/m$^3$ for Liquefied Petroleum Gas and 29.31 MJ/kg for coal, and $I_{ei}$ is the carbon emission factor of fuel $i$. IPCC estimates of $I_{ei}$ were used, and the carbon containment parameter was set to 27.63 gC/GJ. Carbon dioxide is generated during the production of heat and electricity but not during their use.

The carbon emission factors for heat and electricity were determined using the data for thermal power generation and heat supply from the Intermediate Inputs and Transform table in the Energy Balance Sheet. Then, we calculated indirect $CO_2$ emissions from thermal power generation and the production of heat and electricity. The average emission coefficient of the regional power plants according to the Guidelines for Provincial Greenhouse Gas Inventories was used to calculate $CO_2$ emissions from electricity ($C_{electricity}$):

$$C_{electricity} = W_{electricity} \times W_{coal} \times G_{coal} \times I_{coal} \tag{4}$$

where $W_{electricity}$ is the electricity consumption (kWh), $W_{coal}$ is the amount of coal used for electricity production (kg/kWh), $G_{coal}$ is the coal energy transfer coefficient (MJ/Tt), and $I_{coal}$ is the coal carbon containment coefficient (kg C/GJ), set as 0.928 kg/kWh for the East China area.

## 3. Methods
### 3.1. Two-Dimensional GSC Index

A two-dimensional (2D) GSC index was developed, which follows Newton's Law of Gravitation and the degree of compactness proposed by Thinh et al [29]. The study area was divided into grid cells. The area occupied by green space in each cell, the geometric

distance between any two cells, and the total number of cells grids were used to derive the 2D green space compactness index as Equation (5):

$$CI = \frac{\sum \frac{1}{c} \frac{Z_i Z_j}{d^2(i,j)}}{N(N-1)/2} \tag{5}$$

where $CI$ is the 2D GSC index, which is the average power of attraction between the green spaces in grid cells $i$ and $j$; $Z_i$ is the area of green space in cell $i$ (colored in green in cell $i$ in Figure 1a) and $Z_j$ is the area of green space in cell $j$ ($i \neq j$); d ($i, j$) is the geometric distance between cells $i$ and $j$; $c$ is a constant; $N$ is the total number of cells in the grid.

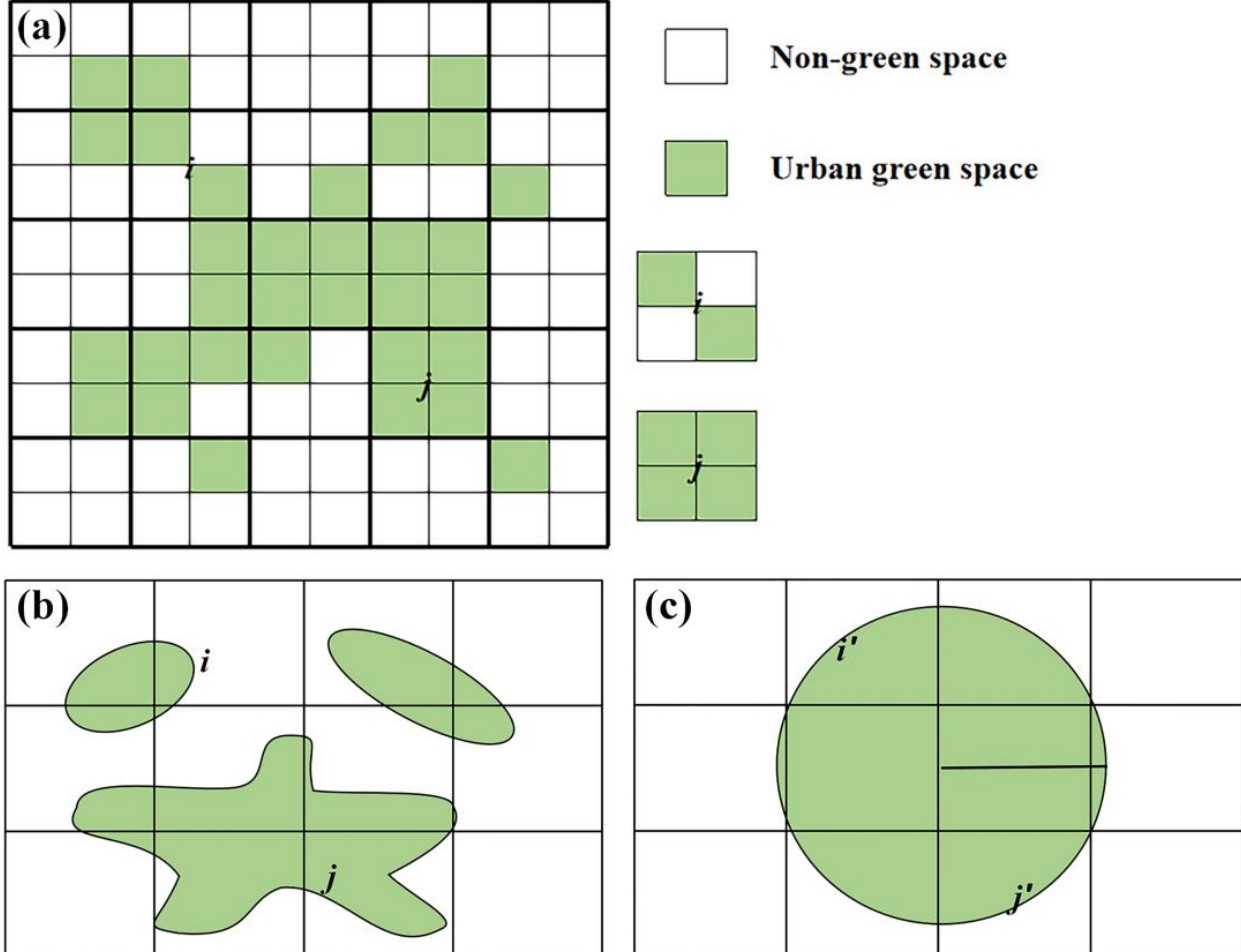

**Figure 1.** Green space compactness model construction. (**a**) Schematic diagram of the grid dividing principal in green space compactness computation model. 10 m resolution global surface cover product—FROM-GLC10 is taken as an example. The pixel size is 10 m × 10 m and the grid size is 20 m × 20 m. $Z_i$ and $Z_j$ are the area (green part) of green space in grid $i$ and $j$; (**b**) Distribution of real urban green space of schematic diagram of the grid dividing principal in the NCI model; (**c**) Illustration of the equivalent circle of schematic diagram of the grid dividing principal in the NCI model.

Differences in green space scales can cause errors in CI. In order to eliminate this difference and facilitate the comparison of CI at different green space scales, the CI is normalized to the NCI, which is the constructed green space compactness index. In the normalized green space compactness index model (Figure 1b,c), the circle with the same area as the real green space is assumed to be the most compact green form, and the

normalized green space compactness index is obtained by comparing CI with the most compact green space form. The calculation formula is as follows:

$$NCI = \frac{CI}{CI_{\max}} = \frac{M(M-1)}{N(N-1)} \times \frac{\sum\limits_{i=1}^{n}\sum\limits_{j=1}^{n}\frac{Z_i Z_j}{d^2(i,j)}}{\sum\limits_{i'=1}^{n}\sum\limits_{j'=1}^{n}\frac{S_i' S_j'}{d'^2(i',j')}} \tag{6}$$

where *CI* and *CI<sub>max</sub>* are the compactness of green space calculated by Equation (5) and the compactness of the equivalent circle with the same green space area, respectively. $S_i'$ and $S_j'$ are the areas of green areas in grids $i'$ and $j'$ in the equivalent circles, $d'(i', j')$ is the geometric distance between grids $i'$ and $j'$, and *M* is the total number of grids in the equivalent circles.

The microclimate is generally defined as the local climate within 1 km² around a building, represented by meteorological factors such as air humidity, solar radiation, and wind speed in some previous studies on building energy consumption [30]. In this study, GSC was used as the indicator of the microclimate environment. The aim was to find the most effective impact distance of the GSC on the BECCE within 1000 m of the building. The GSC index of the different buffers needed to be acquired. The People's Bank building in each city was placed at the center of a grid, and a first boundary was drawn at a distance of 50 m away from the building. Subsequent boundaries were drawn at increments of 50 m away from the building, resulting in twenty boundaries from 20 buffer zones that extend a total of 1 km. Figure 2a is a schematic diagram of the green space buffer zone within 1000m of the People's Bank of Xiamen. The NCI was calculated for each buffer zone (Figure 2b). Values of NCI range between 0 and 1, and a higher NCI is associated with a higher level of connectivity between green spaces.

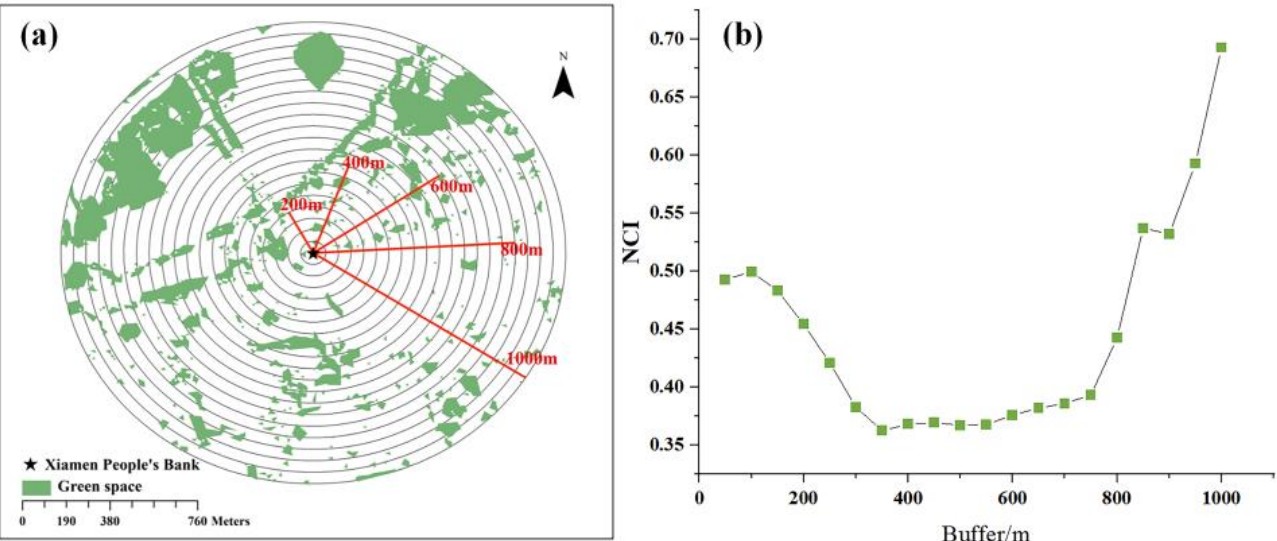

**Figure 2.** Schematic of the green space compactness for different buffer zones. (**a**) Xiamen People's Bank of China is taken as an example. The green space within 1 km of the People's Bank of Xiamen is divided into 20 buffer zones; (**b**) The value of green space compactness of 20 buffer zones. (Note: NCI is the value of green space compactness).

*3.2. Three Models*

3.2.1. Modeling Specification/Purpose of Using the Model

BECCE was used as the dependent variable (Y). Energy consumption membership (X₁), the building's square foot area (X₂), Gross Domestic Product per person (X₃), education spending (X₄), heating degree days (X₅), cooling degree days (X₆), and NCI (X₇) were used as the independent variables (X). The PLSR was performed using the software SIMCA-P. The RF model was constructed by software R. Geographical detector was run in EXCEL.

The variable importance of projection (VIP) of PLS is an important parameter to reflect the importance of a predictor for both the independent and the dependent variables. Therefore, the values of VIP were used to assess the most effective distance of GSC on BECCE. RF can measure the relative importance of variables to the classification during the classification process. Therefore, it was used to assess the importance of GSC in different buffer zones on BECCE. The q value in the factor detector was used to probe the role of GSC at different spatial scales in driving BECCE. Based on the 20 sets of buffers, we constructed 20 sets of models. The variable importance of projection (VIP) of PLS, the importance value of RF, and the q value of GDM were used to indicate the contribution of the green space compactness to BECCE. We could obtain 20 groups of the contribution of GSC on BECCE. The radius of the buffer with the highest contribution value of GSC on BECCE is the most effective influence distance of GSC on BECCE. When there is multicollinearity between the selected variables, PLS is a more suitable method, the RF model can be chosen to calculate when the amount of research data is large, and GDM can be used when there is an obvious spatial relationship between the research objects. We added a table containing information about each variable in Appendix A.

### 3.2.2. Partial Least Squares Regression (PLSR)

PLS regression is a multivariate statistical method with wide applicability [31], which was first proposed by Wold and Albano [32] in 1983. It builds a linear regression model via data dimension reduction, information synthesis, and screening technology to extract new comprehensive components with optimal interpretation of the system [33]. To evaluate the influence of GSC in different buffer zones on BECCE, we constructed the PLS model to explore the relationship between a set of predictor data—X ($n \times m$)—and a response vector—Y ($n \times l$). The variable importance of projection (VIP) and regression coefficients (RCs) are the two important parameters to reflect the importance of a predictor for both the independent and the dependent variables [34]. The PLSR models were established in Simca-p software. Data needed to be normalized before running PLSR.

### 3.2.3. Random Forests (RF)

RF was invented by Breiman [35] as an ensemble learning method that can be used for classification, regression, clustering, interaction detection, and variable selection [36]. In this method, multiple sub-samples are extracted from the original sample, and a decision tree is used to model each sample; the predictions from multiple decision trees are combined to obtain the final prediction by voting. RF can measure the relative importance of variables to the classification during the classification process [37].

### 3.2.4. Geographical Detector Model (GDM)

The geographical detector model is a spatial analysis model based on spatial differentiation. It uses factor power as a measure to effectively detect and identify correlations between geographic attribute and its explanatory factor [38]. The GDM can quantify the explanatory power of a single factor (X) with respect to the dependent variable (Y), making it possible to rapidly and accurately detect correlations between spatial elements. It has been widely used in many areas of environmental research and is available from (www.geodetector.org). The factor detector was used to probe the role of GSC at different spatial scales in driving BECCE [39]. The geographical detector assumes that the spatial distribution of the dependent variable is similar to that of a factor that influences the dependent variable, and is given by Equation (7):

$$q = 1 - \frac{1}{N\sigma^2} \sum_{h=1}^{L} N_h \sigma_h^2 \tag{7}$$

where $q$ indicates the explanatory power or relative importance of X with respect to Y, or the degree of the spatial stratified heterogeneity of Y; $q$ ranges from 0 to 1; $q = 0$ indicates

that Y has no spatial stratified heterogeneity or there is no association between Y and X; $q = 1$ indicates that Y has maximum spatial stratified heterogeneity or Y is fully determined by X; $N$ and $N_h$ indicate the total number of zones and the number of zones in stratum $h$, respectively; $\sigma^2$ and $\sigma_h^2$ are the variances of all statistical units within the entire study area and within stratum $h$, respectively. The study area is divided into $L$ layers, and $h$ ranges between 1 and $L$ (h = 1, 2, . . . , L).

### 3.3. Hot and Cold Spot Analysis

The hot and cold spot statistic has been developed to detect hot and cold spots on maps [40]. It is a local spatial autocorrelation statistic that assumes that the spatial associations are locally heterogeneous, even if global spatial autocorrelation exists [41]. The hot and cold spot analysis in ArcGIS is a statistical tool. By calculating the Z value and P value of each element in the dataset, it obtains the spatial clustering position of the high-value or low-value elements, so that it can intuitively understand where the high-value or low-value elements are clustered and the degree of aggregation [42]. The hot and cold spot analysis tool was used to detect the high and low values and the degree of clustering of BECCE in different regions [43].

## 4. Results

### 4.1. Socioeconomic Conditions and Building Characteristics

The floor area of the buildings and the number of energy consumption memberships are the two basic building characteristics that determine energy use in a building. For the 59 buildings that were examined in this study, the number of building energy users ranged from 114 in Sanya City to 1648 in Shanghai. The largest building was in Changsha City (90,000 m$^2$), and the smallest was in Zhongshan City (6200 m$^2$). Buildings in provincial capitals were generally larger and had more energy consumption membership. Gross Domestic Product per person and education spending were used as indicators of socioeconomic conditions. Gross Domestic Product per person was the highest in Erdos City and the lowest in Shantou City; the average was 89,700 yuan. Education spending ranged from 10,406 to 2,827,117 yuan (Table 1), with Shanghai and Beijing investing far more on education than other cities.

**Table 1.** Socioeconomic conditions and building characteristics.

|  | Max | Min | Mean |
| --- | --- | --- | --- |
| ECM/person | 1648 | 114 | 550 |
| FA/hm$^2$ | 9 | 0.62 | 3.28 |
| GRP/yuan | 153,110 | 31,284 | 89,700 |
| ES/yuan | 2,827,117 | 10,406 | 293,778 |

### 4.2. Regional Climate

Heating degree days and cooling degree days are selected for indicating regional climate. High heating degree days is mainly exhibited in the northern or northwestern China with the highest heating degree days 5157.24 °C·day, which is 56.19 times greater than the smallest heating degree days 91.79 °C·day observed in southern China (Figure 3b). In contrast to the heating degree days, the high values of cooling degree days are mainly concentrated in southern China (Figure 3a). Among the selected cities, Harbin has the highest heating degree days and Zhuhai has the highest cooling degree days of 5157.2 °C·day and 472.7 °C·day, respectively.

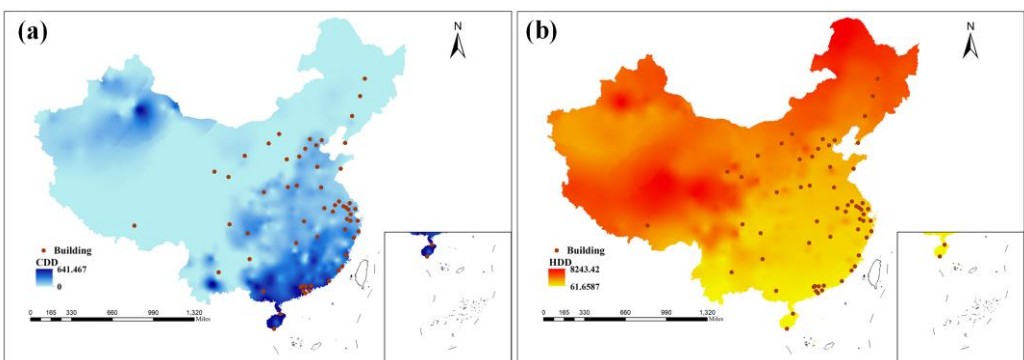

**Figure 3.** Regional climate values for the selected buildings. (**a**) CDD of selected building locations; (**b**) HDD of selected building locations. (Note: CDD represents cooling degree days. HDD represents heating degree days. The unit of CDD and HDD is °C·day).

*4.3. BECCE*

The BECCE of the People's Bank of China in the 59 selected cities is mainly between 250 and 4000 t. The significant differences are obviously exhibited in BECCE among regions. BECCE was generally higher in the north. Taiyuan (9,101,392 tons of $CO_2$), Changchun (6,236,725 tons of $CO_2$), and Xian (6,111,328 tons of $CO_2$) had the three highest BECCE. In the south, BECCE was lower; BECCE was the lowest in Sanya City (255,317 tons of $CO_2$), as shown in Figure 4.

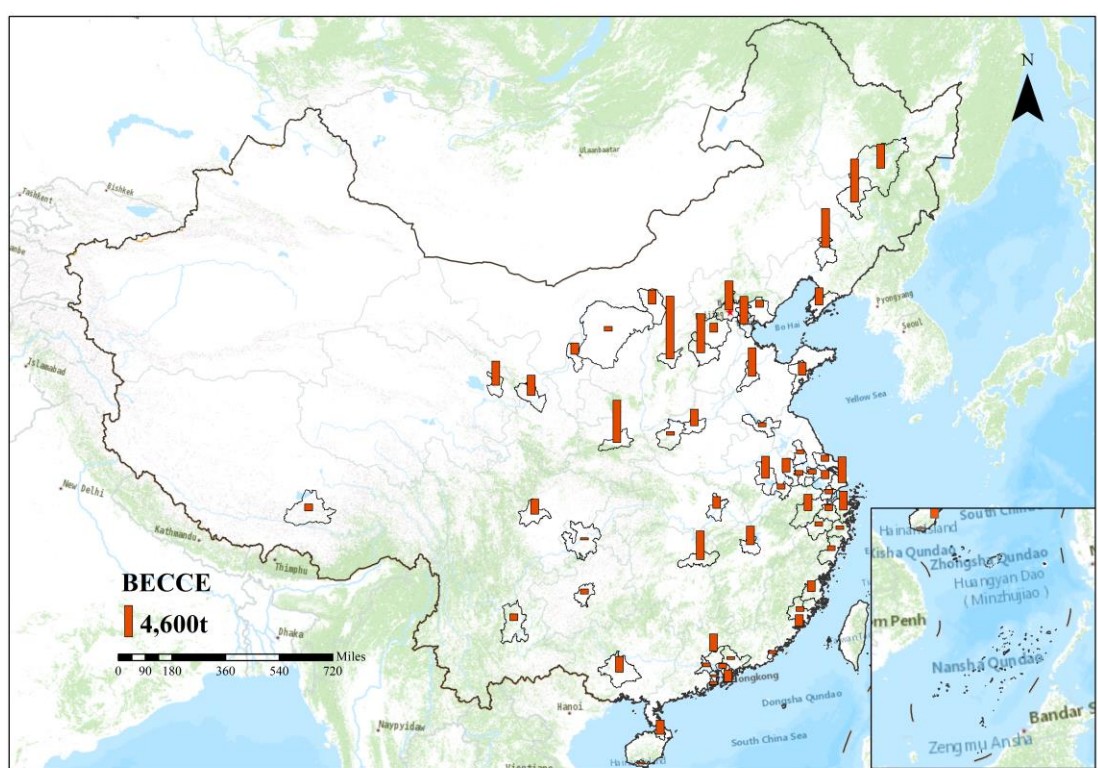

**Figure 4.** The building energy consumption carbon emission of the People's Bank of China in the 59 selected cities. (Note: The unit of building energy consumption carbon emission is tons).

*4.4. NCI*

NCI varied across the 59 cities and 20 buffer zones. It varied little with distance. Taking into consideration all 20 buffer distances, NCI was the highest in Sanya City and the lowest in Wuxi City. The NCI of the People's Bank of Sanya fluctuates between 0.72 and 0.89, while the NCI of the People's Bank of Wuxi only fluctuates between 0.17 and 0.25.

### 4.5. Validation of Three Model

As shown in Table 2, we found that the variance inflation factor (VIF) value of ECM is higher than five, therefore, it can be considered to have multicollinearity among the independent variables.

**Table 2.** Collinearity diagnostics for the PLSR of 250 m buffer zone.

| Independent Variable | Tolerance | VIF |
| --- | --- | --- |
| ECM | 0.177 | 5.651 |
| FA | 0.262 | 3.812 |
| GRP | 0.815 | 1.228 |
| ES | 0.442 | 2.263 |
| HDD | 0.305 | 3.276 |
| CDD | 0.322 | 3.106 |
| NCI | 0.845 | 1.183 |

We verified the PLS model reliability using response permutation. The standards to verify the effectiveness of the model were: all blue $R^2$ values on the left were lower than the initial values on the right, or the blue regression line of $Q^2$ crossed with the vertical axis (left) or was lower than the abscissa. The $R^2$ value indicates the feasibility level of the model. When all green $R^2$ values on the left were lower than the initial values on the right, the original model was proven effective. As shown in Figure 5a, the model is valid. We have also supplied a regression equation result for all 20 models in Appendix B.

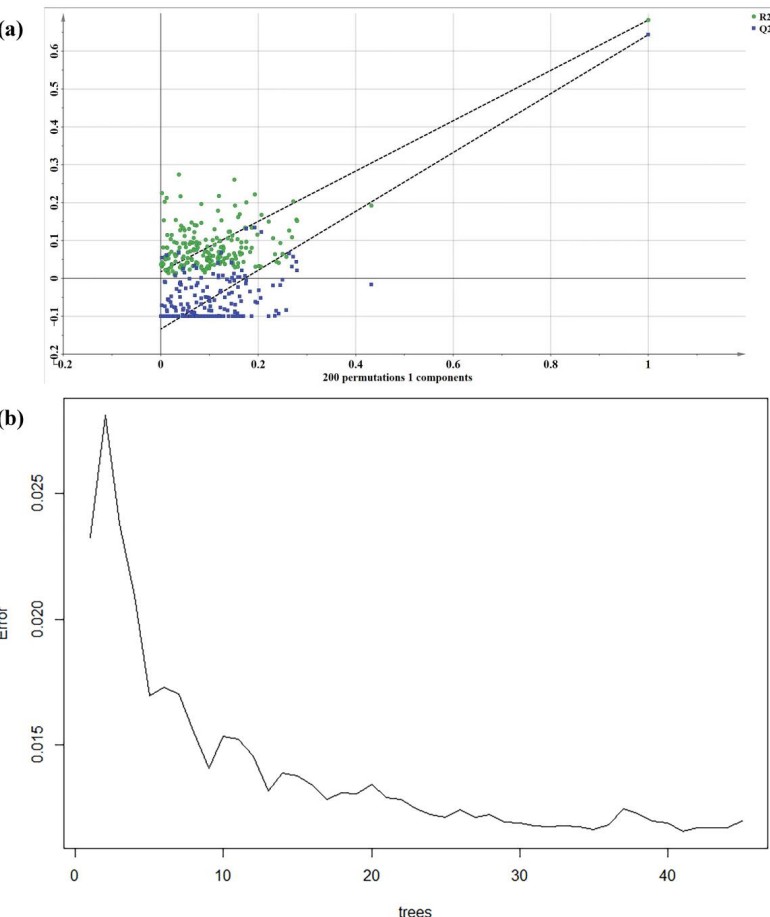

**Figure 5.** Validation of models. (**a**) Permutation test of partial least squares regression model. Criteria for model validation: all the $R^2$ and $Q^2$ are smaller than the original values according to the sequence

distribution of the partial least squares regression model for the 250 m buffer zone; (**b**) Generalization error and number of regression trees of the random forest model. The number of regression trees corresponding to the minimum error value is selected.

The accuracy of the RF model is affected by the number of regression trees (NTREE). To determine the optimal NTREE value, we computed the generalization errors of the RF model for NTREE values of 59. As shown in Figure 5b, generalization error gradually decreased as NTREE increased; at the NTREE value of approximately 40, the generalization error of the model started to converge, which indicates that the model does not over fit. Therefore, in this study, we used the NTREE value of 40.

The $q$ value of the factor detector in the geographic detector represents the explanatory power of the independent variable NCI on the dependent variable BECCE, and the larger the $q$ value, the greater the explanatory power. However, the p value represents the significance level of independent variables. If $p < 0.01$, it is considered to be extremely significant. The $p$ value of NCI of 20 groups of independent variables in this study is all $<0.01$, which proves that the factor detector is reliable.

### 4.6. The Most Effective Influence Distance of GSC on BECCE

The SIMCA-P data analysis software was used to conduct PLSR. BECCE was the dependent variable, and the seven independent variables were the number of energy consumption membership, the building's square foot area, Gross Domestic Product per person, education spending, heating degree days, cooling degree days, and NCI. Figure 6 shows the contribution of NCI to BECCE for the 20 buffer zones. The contribution varied from 0.04 to 0.16; the top three values were 0.142, 0.15, and 0.16 and were in the 750 m, 350 m, and 250 m buffer zones, respectively. The largest contribution of GSC to BECCE was in the 250 m buffer zone.

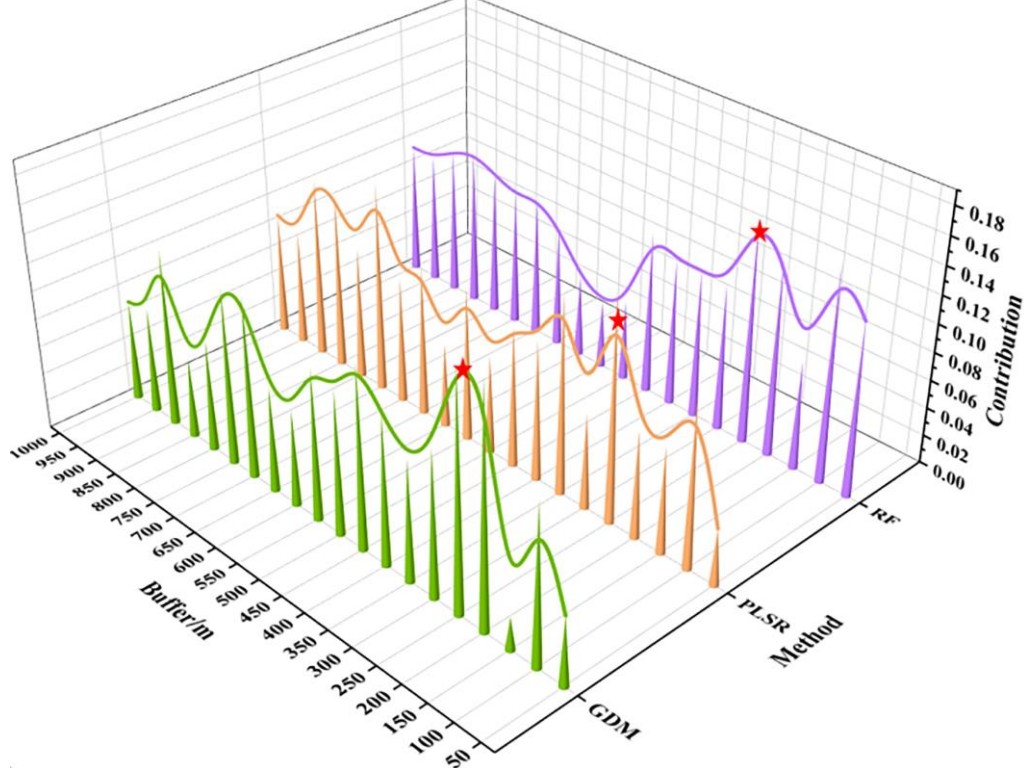

**Figure 6.** Contribution of GSC to BECCE in different buffers. The height of the green cone represents the contribution of GSC to BECCE in the different buffers obtained by GDM; the height of the orange

cone represents the contribution of GSC to BECCE in the different buffers obtained by PLSR; the height of the purple cone represents the contribution of GSC to BECCE in the different buffers obtained by RF. (Note: GSC represents green space compactness. BECCE represents building energy consumption carbon emission. GDM represents geographical detector model. PLSR represents partial least squares regression. RF represents random forests).

The importance of the independent variables to the predicted values was assessed for each model using different indices. High index values are associated with a large importance from the variable. Figure 6 shows that the importance of NCI to BECCE varied across the 20 buffer zones and was between 0.04 and 0.16. Importance was around 10% for most zones and was the highest in the 250 m buffer zone. Results from the RF model also indicate the highest importance of GSC to BECCE in the 250 m buffer zone.

Factor detection was used to estimate the degree to which GSC can explain BECCE. The degree of explanation varied between 0.023 and 0.168 across the 20 buffer zones. From the 50 m to the 250 m buffer zone, there were large fluctuations in the degree of explanation; the lowest value was in the 150 m buffer zone, and the highest value was in the 250 m buffer zone. From the 300 m to the 1000 m buffer zone, variations were relatively small (0.07–0.12). The degree of explanation was the highest for the 250 m buffer zone. Results from the GDM also indicate that GSC has the largest effect on BECCE in the 250 m buffer zone (Figure 6).

*4.7. Optimization of Green Space Compactness in the 250 m Buffer Zone for Emission Reductions*
4.7.1. Contribution of Factors to BECCE

Figure 7 shows the contribution of the independent variables to BECCE for the 250 m buffer zone according to PLSR, RF, and GDM. Building characteristics, such as the number of energy consumption membership and the building's square foot area, had the highest impact on BECCE. Regional climate, represented by heating degree days and cooling degree days, had the second-highest impact. GSC, which is an indicator of microclimate, had the third-highest impact. Socioeconomic conditions, as indicated by Gross Domestic Product per person and education spending, were the smallest contributor to BECCE.

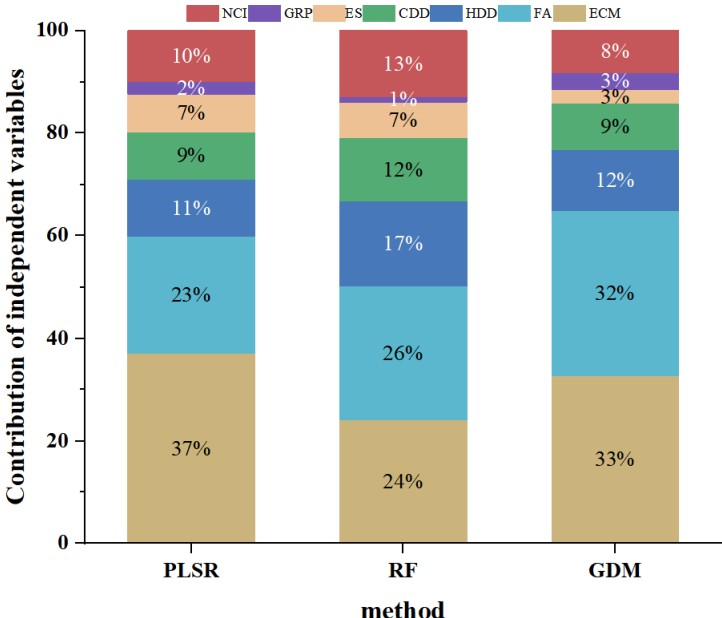

**Figure 7.** Contribution of independent variables to building energy consumption carbon emission obtained by PLSR, RF, GDM. (Note: NCI was the value of green space compactness. GRP represents Gross Domestic Product per person. ES represents education spending. CDD represents cooling

degree days. HDD represents heating degree days. FA represents the building's square foot area. ECM represents energy consumption membership. The unit of GRP and ES is yuan, the unit of CDD and HDD is °C·day, the unit of FA is hm$^2$, the unit of ECM is person. PLSR represents partial least squares regression. RF represents random forests. GDM represents geographical detector model).

### 4.7.2. Impacts of Factors on BECCE

The software SIMCA-P gave the following PLSR model for the independent and dependent variables:

$$y(BECCE) = 0.0716 + 0.285 \times ECM + 0.260 \times FA - 0.015 \times GRP + 0.097 \times ES + 0.180 \times HDD - 0.101 \times CDD - 0.159 \times NCI$$
$$(R^2 = 0.68) \tag{8}$$

PLSR is suitable to deal with multiple linear regression with more environmental factor variables than sample size. Obviously, most of the variables were significantly correlated with each other ($p < 0.05$). Equation (8) showed that the number of energy consumption memberships, the building's square foot area, heating degree days, education spending were positively correlated with BECCE; the correlation coefficients of the number of energy consumption memberships and the building's square foot area were higher than those of education spending and heating degree days. Cooling degree days, Gross Domestic Product per person, and NCI were negatively correlated with BECCE; the magnitude of the correlation coefficient of Gross Domestic Product per person was the lowest, and that of NCI was the highest. High NCI is associated with low BECCE.

### 4.7.3. Optimization of GSC for BECCE Reduction

GSC negatively impacts energy use, and therefore GSC can be improved to reduce BECCE. Figure 8a shows that hot and cold spot analysis was conducted for the 250 m buffer zone, and results indicate spatial heterogeneity. We divided BECCE into five classes by cold/hot spot analysis using the natural breakpoint method. The natural break-point method is a statistical method of classification according to the law of numerical statistical distribution, which maximizes the differences between classes and then divides the studied objects into groups with similar properties. BECCE of the 250m buffer zone is divided into (I) hot spots (very high BECCE, 2947~9101 tons), (II) secondary hot spots (high BECCE, 2174~2710 tons), (III) transition area (BEECE was 1009~2057 tons), (IV) secondary cold spots (low BECCE, 669~996 tons), and (V) cold spots (very low BECCE, 225~683 tons). Hot spots were in the north and center, and cold spots were in the south and southwest (Figure 8a). For cold spots, BECCE was 255–683 tons and GSC was 0.44–0.85. Emissions were the lowest at the central office of the People's Bank in Sanya City (255 tons). For hot spots, BECCE was 2947–9101 tons, and GSC was 0.18–0.55. Emissions were the highest at the central branch of the bank in Taiyuan.

Drawing on the study of Wu [39], we adjusted the GSC from the mean of one interval to the other to reduce BECCE. We analyzed different scenarios of BECCE reduction and found that an increase in GSC from 0.39 (mean GSC of hot spots) to 0.56 (mean GSC of cold spots) would lead to a decrease of 4189 tons in BECCE (Figure 8b). Replacing class I GSC with class II GSC (I–II), class III GSC (I–III), and class IV GSC (I–IV) reduced BECCE by 2259, 3096, and 3845 tons, respectively. Replacing class II GSC with class III GSC (II–III), class IV GSC (II–IV), and class V GSC (II–V) reduced BECCE by 837, 1586, and 1930 tons, respectively. Substituting class III GSC with class IV GSC (III–IV), and class V GSC (III–V) reduced BECCE by 749,1093 tons, respectively. Replacing class IV GSC by class V GSC (IV–V) reduced BECCE by 344 tons. The largest carbon reduction resulted from replacing class I with class V GSC; the smallest reduction resulted from replacing class IV with class V GSC.

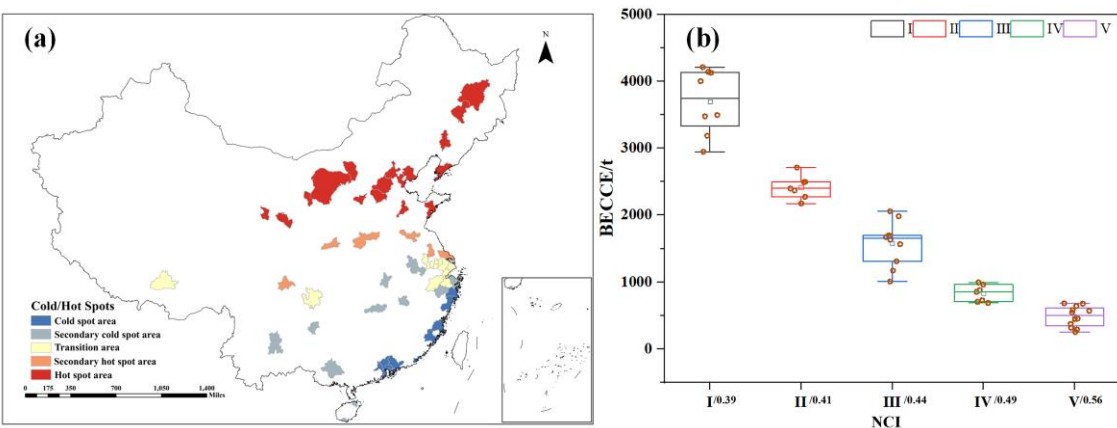

**Figure 8.** Optimization of the green space compactness. (**a**) Distribution of hot and cold spots of BECCE; (**b**) The relationship between green space compactness and BECCE. Reductions in BECCE are associated with the replacement of GSC of one category with that of another category. (Note: I/0.39, II/0.41, III/0.44, IV/0.49, V/0.56 represented mean of the NCI corresponding to the hot spot, secondary hot spots, transition area, secondary cold spots, and cold spots, respectively. GSC represents green space compactness. NCI was the value of green space compactness. BECCE represents building energy consumption carbon emission. The unit of BECCE is ton).

## 5. Discussion

In this study, three different methods were used to obtain the most effective distance for GSC to have an impact on BECCE. Through the cross-validation of the three methods, we identified the most effective distance for GSC to influence BECCE at 250 m from the building.

### 5.1. The Significance of the Most Effective Distance

Green spaces can affect the urban environment at different scales. In this study, we found that 250 m is the distance at which GSC has the most effective impact on BECCE, which provides a new cost-effective option to reduce carbon by improving green spaces. The influence of GSC on BECCE was the highest for the 250 m buffer zone. GSC also contributed to BECCE reduction at other buffer distances, such as 750 m. However, considering the costs associated with the construction of green spaces, we conclude that the buffer distance of 250 m is the most cost-effective option. Therefore, future research can focus on the impact of GSC on BECCE within 250 m of the building, which can improve the efficiency of transforming the green space to reduce carbon. The "transpiration" of green spaces can effectively regulate the local thermal environment around them, making them suitable for use as transformation measures for cold islands [44]. In a previous study, the relationship between the cooling effect intensity (PCI) and the buffer distance of the selected park has been determined. The results showed that the park cooling intensity for the parks has varying intensity values in each buffer zone. The PCI intensity values increased sharply between 50 and 200 m buffer zones at the park from 0.93 °C to 1.88 °C. The intensity value decreased from 1.76 °C to 1.60 °C between 250 m and 300 m, respectively, while between 350 m and 450 m distance, there was a reduction in PCI intensity from 1.66 °C to 1.59 °C. The results of this study may provide an explanatory basis for our findings [45]. Our study shows that a better configuration of green spaces can lead to considerable reductions in BECCE. Apart from energy, the cooling effect of green space can also be used to offset the adverse effect of climate change [46] and urban warming [47] on the health of urban habitat. Numerical models have been widely used to study the impact of green space on urban climate and heat islands. For instance, WRF-UCM model was used to study the water effects of urban heat islands in summer [48], while Envi-met was used to assess the effect of green space on the wind environment and to analyze the effect of street greenbelts on microclimate in an area of China using a numerical simulation method [49,50]. The

WRF-UCM model is based on a region, and the Envi-met simulation also has to set the boundaries. Therefore, the most effective distances obtained in this study can provide a reference to the range of these simulations.

### 5.2. Optimization of Green Space Configuration Based on the Most Effective Distance

The contribution of microclimate to BECCE was approximately 10%, which is highly consistent with previous studies [51–53]. Green spaces impact microclimate, which indirectly influences BECCE, and BECCE reduction varies with the spatial distribution of green spaces. Our results show that the contribution of green space to BECCE reduction varies with the spatial configuration of the green spaces. The ecological function of urban green spaces is dependent on spatial configuration. Some studies show that the cooling effects of urban green spaces are dependent on the size, shape, and type of green space, and that green space configuration modifies temperature, wind, and humidity in urban environments [54,55]. GSC influences microclimate and considerably impacts carbon emissions. We examined building characteristics, regional climate, socioeconomic conditions, and the configuration of green spaces. We found a significant spatial heterogeneity of BECCE within the study area, and GSC also ranged within the different rank regions. These results can be used to optimize emission reductions. In our study, GSC was the highest for the Sanya City central branch building for the 250 m buffer zone and lowest for the building in Taiyuan City, as shown in Figure 9. Replacing Taiyuan's GSC with Sanya's GSC would likely result in considerable emission reductions. A single green space compact form perspective without considering other factors to reduce BECCE may still lack scientific rigor. We will further explore the change law of compactness in long-time scale, coupling the driving factors of BECCE, and explore the suitable spatial form for transformation to provide a more scientific strategy to support building carbon reduction.

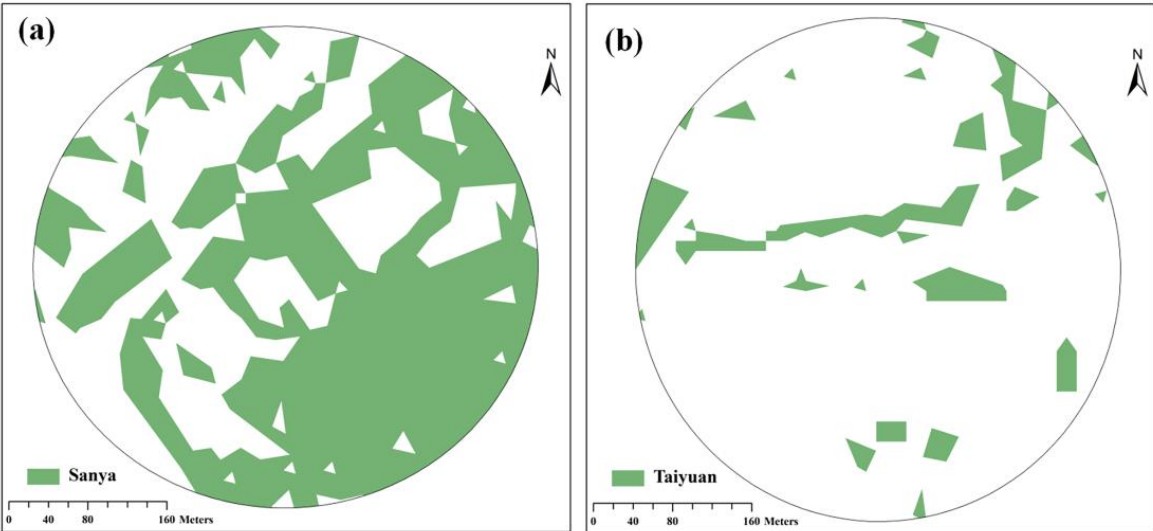

**Figure 9.** The transformation of two green space compactness types. (**a**) The distribution of green spaces within 250 m of the People's Bank of Sanya; (**b**) The distribution of green spaces within 250 m of the People's Bank of Taiyuan.

### 5.3. Limitations of the Study

Different driving factors have different effects on BECCE, so changing the green space form to a better energy-saving building may result in another factor changing to a bad energy-saving building. Therefore, it is necessary to analyze the integrated effect of the integrated elements on building energy savings.

The subjects of this study are distributed throughout China, with a complex natural environment. Other factors, such as population density around buildings, the number of

nearby buildings, and building age, also affect the BECCE. However, due to data availability, these factors were not taken into account in the study.

In the future, if the data are available, we will consider the most effective impact distance of compact green space on building energy consumption carbon emissions worldwide to obtain a universal rule and provide more efficient theoretical support for carbon reduction.

## 6. Conclusions

In the context of rapid urbanization, the continued growth of BECCE limits the sustainable development of cities. This study proposes a nature-based mitigation strategy. We used the green space within 1 km of the People's Banks in 59 cities across China and developed green space compactness index to represent the micro-climate environmental characteristics affecting BECCE. We obtained the contribution of GSC to building energy consumption carbon emission in 20 groups of buffers using partial least squares, random forest, and geographic detectors. A spatial range of 250 m can be used to consider the optimal influence distance of GSC on BECCE. In future studies, when studying the effect of GSC on BECCE, we can focus on the spatial range of 250 m.

Based on this conclusion, we divided the BEECE into five categories based on hot and cold spots analysis. We optimized green space configuration for reductions in building emissions by exploring different BECCE levels and their related GSC. The optimization of green space configuration can improve the potential for carbon reduction. To reduce emissions from 4675 to 486 tons, GSC needs to increase from 0.39 to 0.56. These conclusions can provide benchmark information for landscape planning and building energy conservation in the future and can be used to develop more effective measures to reduce emissions to respond to China's strategic plan to reach peak $CO_2$ emissions by 2030 and be carbon neutral by 2060.

Due to the availability of data, the selected cities are not evenly distributed. When the study conditions are suitable, more cities should be selected to obtain more generalized findings. Green spaces reduce temperature in urban environments by mitigating the heat island effect. This process involves many parameters of green spaces, including compactness, location, and the diversity of plants. Therefore, to further explore the influence of green space on BECCE, other two- and three-dimensional green space indices should be explored.

**Author Contributions:** Conceptualization, R.J. and H.Y. (Hong Ye); data curation, R.J., M.Z. and X.W.; methodology, R.J., H.Y. (Hong Ye), M.Z. and H.Y. (Han Yan); software, H.Y. (Han Yan) and Z.Z.; visualization, R.J. and Y.B.; writing—original draft preparation, R.J., Y.Z. and K.W.; formal analysis, R.J. and H.Y. (Hong Ye); writing—review and editing, R.J., H.Y. (Hong Ye), K.W., Y.Z. and Y.B.; resources, H.Y. (Hong Ye); project administration, H.Y. (Hong Ye); funding acquisition, H.Y. All authors have read and agreed to the published version of the manuscript.

**Funding:** This research was funded by the International Partnership Program of the Chinese Academy of Sciences, grant number 132c35kysb2020007 and the Ningbo Commonweal Science and Technology Planning Project, grant number 2021S081.

**Data Availability Statement:** The data presented in this study are available upon request from the corresponding author.

**Conflicts of Interest:** The authors declare no conflict of interest.

## Abbreviations

| | |
|---|---|
| BECCE | Building energy consumption carbon emission |
| GSC | Green space compactness |
| IPCC | Intergovernmental Panel on Climate Change |
| UHI | Urban heat island |
| PLS | Partial least squares |
| RF | Random Forest |
| GDM | Geographical detector model |

| | | | |
|---|---|---|---|
| GRP | Gross Domestic Product per person | | |
| ES | Education spending | | |
| CDD | Cooling degree days | | |
| HDD | Heating degree days | | |
| PLSR | Partial Least Squares Regression | | |
| VIP | Variable importance of projection | | |
| RCs | Regression coefficients | | |
| ECM | Energy consumption membership | | |
| FA | Floor area of the buildings | | |
| VIF | Variance inflation factor | | |
| NTREE | Number of regression trees | | |
| PCI | Cooling effect intensity | | |

## Appendix A

**Table A1.** Information with each variable after standardization.

| BECCE | ECM | FA | GRP | ES | HDD | CDD | GSC |
|---|---|---|---|---|---|---|---|
| 0.318189 | 0.747646 | 0.772077 | 0.612256 | 0.197687 | 0.069528 | 0.816938 | 0.190194 |
| 0.134185 | 0.167608 | 0.091641 | 0.060579 | 0.006136 | 0.47575 | 0.167753 | 0.359675 |
| 0.449241 | 0.827997 | 0.392144 | 0.458169 | 1 | 0.465784 | 0.18445 | 0.47881 |
| 0.074628 | 0.127433 | 0.135465 | 0.482123 | 0.075007 | 0.275049 | 0.227713 | 0.231952 |
| 0.25124 | 0.458883 | 0.531519 | 0.296004 | 0.087959 | 0.242861 | 0.101426 | 0.342488 |
| 0.264145 | 0.330822 | 0.32189 | 0.511964 | 0.150569 | 0.489636 | 0.02643 | 0.532644 |
| 0.08447 | 0.103578 | 0.079041 | 1 | 0.00641 | 0.694908 | 0.00258 | 0.329226 |
| 0.180361 | 0.283114 | 0.19932 | 0.295874 | 0.031145 | 0.114525 | 0.648707 | 0.400756 |
| 0.05566 | 0.19774 | 0.177555 | 0.466943 | 0.05258 | 0.065006 | 0.846282 | 0.461438 |
| 0.116549 | 0.351538 | 0.113591 | 0.214851 | 0.041946 | 0.306952 | 0.018555 | 0.590675 |
| 0.437372 | 0.392341 | 0.282942 | 0.208651 | 0.033625 | 0.935144 | 0.0127 | 0.438582 |
| 0.237357 | 0.328311 | 0.626597 | 0.187396 | 0.001776 | 0 | 0.997262 | 0.822529 |
| 0.294817 | 0.338355 | 0.390907 | 0.478823 | 0.183819 | 0.239519 | 0.290859 | 0.530976 |
| 0.372273 | 0.416196 | 0.53381 | 0.283399 | 0.10105 | 0.285853 | 0.225594 | 0.552416 |
| 0.272004 | 0.4457 | 0.297834 | 0.436345 | 0.010848 | 0.741209 | 0 | 0.535206 |
| 0.462519 | 0.500942 | 0.585977 | 0.3611 | 0.032104 | 0.360428 | 0.259573 | 0.404124 |
| 0.076188 | 0.110483 | 0.208484 | 0.314608 | 0.048011 | 0.254038 | 0.255843 | 0.39636 |
| 0.075348 | 0.096673 | 0.160372 | 0.236696 | 0.045519 | 0.206499 | 0.45477 | 0.588682 |
| 0.104947 | 0.284369 | 0.231394 | 0.22144 | 0.042641 | 0.177875 | 0.197884 | 0.319651 |
| 0.138092 | 0.28688 | 0.06644 | 0.223501 | 0.009517 | 0.780833 | 0.088589 | 0.625301 |
| 0.360755 | 0.326428 | 0.271029 | 0.213515 | 0.009293 | 0.703726 | 0.030689 | 0.182128 |
| 0.07246 | 0.152542 | 0.098514 | 0.184551 | 0.028113 | 0.342095 | 0.248606 | 0.393476 |
| 0.327729 | 0.575016 | 0.55672 | 0.297919 | 0.026101 | 0.208433 | 0.480869 | 0.562853 |
| 0.226869 | 0.34751 | 0.4961 | 0.499013 | 0.156556 | 0.282981 | 0.212251 | 0.538968 |
| 0.27207 | 0.413685 | 0.463933 | 0.151475 | 0.023768 | 0.084661 | 0.813339 | 0.445084 |
| 0.095192 | 0.094162 | 0.292496 | 0.336242 | 0.075303 | 0.292474 | 0.186674 | 0.482099 |
| 0.302671 | 0.392341 | 0.588795 | 0.449334 | 0.149844 | 0.229303 | 0.262033 | 0.678564 |
| 0.203962 | 0.24796 | 0.174118 | 0.439391 | 0.093821 | 0.396883 | 0.061395 | 0.286219 |
| 0.07171 | 0.116761 | 0.234831 | 0.286455 | 0.035563 | 0.121206 | 0.607363 | 0.44175 |

**Table A1.** *Cont.*

| BECCE | ECM | FA | GRP | ES | HDD | CDD | GSC |
|---|---|---|---|---|---|---|---|
| 0.024945 | 0.037037 | 0.065294 | 0.212503 | 0.009539 | 0 | 0.997262 | 0.743105 |
| 0.196639 | 0.210295 | 0.306998 | 0.386959 | 0.059709 | 0.077895 | 0.640265 | 0.442587 |
| 0.052234 | 0.104834 | 0.087179 | 0.086005 | 0.005522 | 0.045787 | 0.821615 | 0.43792 |
| 0.415744 | 1 | 0.964523 | 0.443968 | 0.927631 | 0.247362 | 0.205764 | 0.694095 |
| 0.093119 | 0.158192 | 0.162663 | 0.383194 | 0.063493 | 0.244262 | 0.308466 | 0.332113 |
| 0.184264 | 0.322662 | 0.230226 | 0.725955 | 0.333344 | 0.035457 | 0.94567 | 0.408799 |
| 0.069572 | 0.093534 | 0.12257 | 0.299174 | 0.048069 | 0.058557 | 0.842451 | 0.848888 |
| 0.040386 | 0.138732 | 0.096223 | 0.261586 | 0.068132 | 0.060979 | 0.820498 | 0.424405 |
| 0.630392 | 0.375392 | 0.662222 | 0.381463 | 0.092232 | 0.686276 | 0.024143 | 0.218079 |
| 0.646749 | 0.610797 | 0.902666 | 0.182132 | 0.023926 | 0.528524 | 0.146169 | 0 |
| 0.125113 | 0.180163 | 0.261122 | 0.620085 | 0.266714 | 0.251512 | 0.241707 | 0.415426 |
| 0.051237 | 0.101695 | 0.138928 | 0.221289 | 0.032854 | 0.198716 | 0.314952 | 0.370312 |
| 1 | 0.514124 | 0.640343 | 0.236517 | 0.048224 | 0.555907 | 0.013811 | 0.328429 |
| 0.112443 | 0.213434 | 0.162663 | 0.352434 | 0.027001 | 0.514669 | 0.146036 | 0.496153 |
| 0.480345 | 0.561205 | 0.660962 | 0.486494 | 0.384365 | 0.438771 | 0.259779 | 0.417916 |
| 0.071573 | 0.121783 | 0.153029 | 0.172113 | 0.038146 | 0.178507 | 0.405418 | 0.760851 |
| 0.065014 | 0.150659 | 0.223261 | 0.600955 | 0.123807 | 0.259371 | 0.236065 | 0.203245 |
| 0.073543 | 0.123666 | 0.14548 | 0.263653 | 0.119599 | 0.257308 | 0.244642 | 0.269095 |
| 0.187177 | 0.349655 | 0.504162 | 0.447376 | 0.199722 | 0.262423 | 0.283815 | 0.462167 |
| 0.671526 | 0.448839 | 0.531519 | 0.262328 | 0.045884 | 0.369041 | 0.252873 | 0 |
| 0.367658 | 0.323289 | 0.601395 | 0.170188 | 0.004694 | 0.792069 | 0.021731 | 0.466047 |
| 0.056281 | 0.146265 | 0.174118 | 0.229117 | 0.060046 | 0.33095 | 0.241048 | 0.662404 |
| 0.060566 | 0.124294 | 0.187864 | 0.364357 | 0.038146 | 0.282922 | 0.205997 | 0.449239 |
| 0.183302 | 0.334589 | 0.317308 | 0.273948 | 0.011548 | 0.579842 | 0.066659 | 0.297938 |
| 0.690655 | 0.541745 | 0.624306 | 0.300721 | 0.022926 | 0.817749 | 0.011025 | 0.73958 |
| 0.483692 | 0.505336 | 1 | 0.499759 | 0.077674 | 0.21377 | 0.413991 | 0.415791 |
| 0.291484 | 0.837414 | 0.77895 | 0.312082 | 0.049455 | 0.312287 | 0.335889 | 0.274781 |
| 0.028009 | 0.080979 | 0.038948 | 0.396967 | 0.043696 | 0.046337 | 0.947621 | 0.482284 |
| 0.000722 | 0.007533 | 0.01031 | 0.176073 | 0.133341 | 0.181488 | 0.399687 | 0.561651 |
| 0.043754 | 0.098556 | 0.057276 | 0.547658 | 0.042453 | 0.037504 | 1 | 0.577845 |

## Appendix B

**Table A2.** A regression equation result for all 20 models by PLS.

| | ECM | FA | GRP | ES | HDD | CDD | NCI | Cons | Sig |
|---|---|---|---|---|---|---|---|---|---|
| 1000 m | 0.305201 | 0.35986 | 0.166119 | 0.0817256 | 0.228903 | 0.0473396 | 0.0823886 | 0.0603391 | $p < 0.05$ |
| 950 m | 0.304216 | 0.360182 | 0.167533 | 0.0844465 | 0.229248 | 0.04668888 | 0.101299 | 0.0692255 | $p < 0.05$ |
| 900 m | 0.30183 | 0.3572 | 0.17182 | 0.0848071 | 0.229248 | 0.0459239 | 0.0459239 | 0.086017 | $p < 0.05$ |
| 850 m | 0.300175 | 0.353881 | 0.175214 | 0.0826662 | 0.227778 | 0.0466589 | 0.155681 | 0.0982501 | $p < 0.05$ |
| 800 m | 0.301175 | 0.354913 | 0.177489 | 0.0798333 | 0.225889 | 0.0475465 | 0.158625 | 0.100128 | $p < 0.05$ |
| 750 m | 0.30256 | 0.357197 | 0.179124 | 0.079243 | 0.224287 | 0.0472305 | 0.162457 | 0.101145 | $p < 0.05$ |

**Table A2.** *Cont.*

|  | ECM | FA | GRP | ES | HDD | CDD | NCI | Cons | Sig |
|---|---|---|---|---|---|---|---|---|---|
| 700 m | 0.303659 | 0.356833 | 0.185114 | 0.0764475 | 0.221304 | 0.0460905 | 0.17927 | 0.109838 | $p < 0.05$ |
| 650 m | 0.306978 | 0.362106 | 0.190085 | 0.0757981 | 0.219899 | 0.0453604 | 0.153132 | 0.0977835 | $p < 0.05$ |
| 600 m | 0.3109 | 0.370421 | 0.185572 | 0.0791872 | 0.218151 | 0.043282 | 0.102372 | 0.0706033 | $p < 0.05$ |
| 550 m | 0.311217 | 0.374508 | 0.18063 | 0.0849314 | 0.218977 | 0.0412042 | 0.0959552 | 0.0643862 | $p < 0.05$ |
| 500 m | 0.310596 | 0.373387 | 0.178106 | 0.0819165 | 0.217279 | 0.0399675 | 0.106434 | 0.0685193 | $p < 0.05$ |
| 450 m | 0.286967 | 0.262201 | 0.014691 | 0.0978542 | 0.181298 | 0.101819 | 0.0977435 | 0.0661807 | $p < 0.05$ |
| 400 m | 0.288109 | 0.263244 | 0.0147495 | 0.0982436 | 0.182019 | 0.102224 | 0.0877683 | 0.0617522 | $p < 0.05$ |
| 350 m | 0.287101 | 0.262323 | 0.0146978 | 0.0978997 | 0.181382 | 0.101866 | 0.0891686 | 0.06379 | $p < 0.05$ |
| 300 m | 0.286347 | 0.261634 | 0.0146593 | 0.0976427 | 0.180906 | 0.101599 | 0.0875556 | 0.0647297 | $p < 0.05$ |
| 250 m | 0.284522 | 0.259967 | 0.0145658 | 0.0970203 | 0.179753 | 0.100951 | 0.105011 | 0.0715619 | $p < 0.05$ |
| 200 m | 0.301157 | 0.354913 | 0.177489 | 0.0798333 | 0.225889 | 0.0475465 | 0.158625 | 0.100128 | $p < 0.05$ |
| 150 m | 0.293589 | 0.268251 | 0.01503 | 0.100112 | 0.185481 | 0.104168 | 0.03757 | 0.0377408 | $p < 0.05$ |
| 100 m | 0.274614 | 0.250914 | 0.0140586 | 0.0936417 | 0.173493 | 0.0974355 | 0.0522553 | 0.0614491 | $p < 0.05$ |
| 50 m | 0.319747 | 0.38092 | 0.162974 | 0.0754476 | 0.218005 | 0.0411308 | 0.0126272 | 0.0114444 | $p < 0.05$ |

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
