# Peer review of "Green Space Compactness and Configuration to Reduce Carbon Emissions from Energy Use in Buildings"

_remotesensing, doi:10.3390/rs15061502_

Round 1

Reviewer 1 Report

I think the author's cross-disciplinary research has positive scientific significance and guiding significance for urban planning, but I have several concerns that need the author's reply:

(1) Why does the author choose the People's banks in different cities? What are the reasons? What are the advantages of this kind of buildings for the author's research?

(2) What is the key role of climate factor in the author's research, especially in the comprehensive relationship between heat island, climate, carbon emission and green space, what role does climate factor play and what quantitative role does climate factor play? I hope the author can further clarify and answer.

(3) The author rarely mentions the limitations of his research in the paper, so he hopes to add.

(4) At present, some researches on urban climate, green space and heat island all adopt atmospheric numerical model to carry out their work. The author thinks that whether your research has reference significance with this kind of research can be discussed in the discussion.

(5) The author's research also has certain guiding significance for protecting the health of residents living in urban climate. I hope the author pays attention. Some literature for reference

Projections of future temperature-related cardiovascular mortality under climate change, urbanization and population aging in Beijing, China. Environment International 163 (2022) 107231

Evidence for urban -- rural disparity in temperature -- mortality relationships in Zhejiang Province, Evidence for urban -- rural disparity in temperature -- mortality relationships in Zhejiang Province, China. Environmental Health Perspectives. 2019, 127(3): 037001.

Author Response

请参阅附件。

Reviewer 2 Report

The authors have presented an interesting work which is definitely of great value to sustainability researchers. The paper is also well written. I only have a few suggestions:

1. Figure 2 - 'green' should be 'Green'

2. Figure 5 - a frequency distribution plot in the carbon emission in the same figure will help to understand the data.

3. Figure 10 - A table with what each variable is will help with ready reference. This can be added to the Figure.

4. Equation 8 - is there a chance that the relation with variables is non-linear in nature ! Please discuss a bit about it in this section.

5. Can factors like population density, number of buildings nearby, age of buildings etc. play a role !

6. How can this model be used by others at any other location across the world! What resources would other research groups need! Please discuss this in the manuscript.

Reviewer 3 Report

Manuscript: Green space compactness and configuration to reduce carbon emissions from energy use in buildings

By viewing the distribution of grey and green urban space and its impact on the climate at small spatial scale the paper addresses an issue of most actual relevance. For China, it finds that consolidation of green space around buildings is most effective within a radius of 250 meters to combat climate relevant impacts from energy emissions of buildings (BECCE). As a case study, buildings of the People’s Bank branches in 59 major cities are viewed. The database consists of daily temperature time series provided by weather stations nearby the buildings, data on building technology, ground area, energy use and consumption and some socio-economic data from official statistics. In addition to that, usual empirical equivalent parameters of carbon emissions from fuel consumption are used. A Green Space Compactness index based on gravity theory is proposed to see where green space is well distributed and where not. Here, the distribution and shape of green and grey cells of the cities are displayed in grids. Methods comprise the application of partial least squares regression (PLS), a supervised classification procedure (Random Forest classifier), the Geographical Detector Model, all three to look at the strength of predictors of local BECCE and – additionally - the Getis-Ord statistic to detect hot and cold spots.

There is a lot to say about the paper. Most important issues are the following:

(1)There is no visible relation with remote sensing. Authors should check this (perhaps the green and grey cells within cities are detected by certain remote sensing imagery, but there is no information provided on that). The reader cannot recognize the use of aerial or satellite imagery, but just data from ground-based sensors (temperature) and documented or official statistics. The use of temperature measured by the closest weather station is anyway not very precise given the fact that there are other powerful (highly resolved) open access remote sensing databases available, e.g. from NASA (https://sedac.ciesin.columbia.edu/data/set/sdei-high-res-daily-uhe-1983-2016).

(2)The organisation and presentation of the research is not optimal. Sub-sections sometimes start with figures and tables instead of text. This is confusing and uncommon. Methods and data are presented but neither compared nor justified. The reader might like to know more about the reason why PLS is needed (are predictors really multicollinear, were variance inflators checked, why PLS and not ridge regression or PCA+OLS?). Variables are sometimes confounded, e.g. C is used for energy consumption and an electricity coefficient, while W is also used for energy consumption (lines 111-129). The large number of abbreviations make the reading burdensome. By and large, the reader often needs a lot of interpretation creativity to understand what is actually meant. Actually this is the major issue to deplore, since the idea and visible research effort of authors seem to get concealed by the sometimes poor organisation of sections and the overall presentation.

(3)The conclusions are hard to grasp in their policy consequences. What does it practically mean for policy if the study finds that 250 meters is the optimum distance to reduce building carbon emission? This is the core finding, but readers might ask „so what?“

(4)All other points raised are highlighted in the manuscript attached.

Round 2

Reviewer 1 Report

The author answered my question nicely and the paper is ready for publication

Author Response

       We are very grateful for reviewers’ careful review. In this study, we found that 250 m is the distance at which GSC has the most effective impact on BECCE, which provides a new cost-effective option to reduce carbon by improving green space. Focusing on the 250m building range, we explored the optimization path of the green space compactness. Therefore, this study will contribute to the Special Issue “Remote Sensing in Urban Socio-Ecological Systems Monitoring and Assessment” of this journal.

Reviewer 3 Report

Manuscript: Green space compactness and configuration to reduce carbon emissions from energy use in buildings -  #2

The revised version appears considerably better than the original manuscript, and it is impressive how authors have produced this new version within shortest time. The paper definitively merits publication.

It is a matter of editorial decision. To me, the manuscript as such seems publishable in its present shape, however I would still recommend to invest a bit of time to make the paper really attractive.  

(1) It is less a matter of methodological comprehension of the statistical and AI tools applied (as possibly interpreted by authors in their rebuttal letter) but rather the argumentative justification of exactly this combination of tools selected. Here I feel a bit misunderstood. The presentation of all that could still be improved, for example some clarification in the text about the exact role of the cold/hot spot analysis in relation to the three other models applied to estimate the predictors of BECCE etc.. There is also still the unproved assumption of multicollinearity to justify the PLS regression. I would first check the existence of multicollinearity (e.g. if VIF>5) before deciding on a specific method addressing this issue.

Two minor notes: (2) The 20 regression models in the annex should be better organised in a table. Significance of coefficients and constants should be included. (3) the reference given in the rebuttal letter to justify the temperature thresholds (Yang, L.; Yan, H.; Lam, J.C. Thermal comfort and building energy consumption implications—A review. Appl. Energy 2014, 115, 164–173) should be correspondingly included in the manuscript.
